# Constructing benchmark test sets for biological sequence analysis using independent set algorithms

**Samantha Petti**[1], **Sean R. Eddy**[2]*

**1** NSF-Simons Center for the Mathematical and Statistical Analysis of Biology, Harvard University, Cambridge, Massachusetts, United States of America, **2** Howard Hughes Medical Institute; Department of Molecular & Cellular Biology; and John A. Paulson School of Engineering and Applied Sciences, Harvard University, Cambridge, Massachusetts, United States of America

* seaneddy@fas.harvard.edu

## Abstract

Biological sequence families contain many sequences that are very similar to each other because they are related by evolution, so the strategy for splitting data into separate training and test sets is a nontrivial choice in benchmarking sequence analysis methods. A random split is insufficient because it will yield test sequences that are closely related or even identical to training sequences. Adapting ideas from independent set graph algorithms, we describe two new methods for splitting sequence data into dissimilar training and test sets. These algorithms input a sequence family and produce a split in which each test sequence is less than $p$% identical to any individual training sequence. These algorithms successfully split more families than a previous approach, enabling construction of more diverse benchmark datasets.

**Data Availability Statement:** The splitting algorithms are implemented in C and available here: https://github.com/EddyRivasLab/hmmer/tree/develop. To run the algorithms, the following version of EASEL is needed: https://github.com/

## Author summary

Typically, machine learning and statistical inference models are trained on a "training" dataset and evaluated on an separate "test" set. This ensures that the reported performance accurately reflects how well the method would do on previously unseen data. Biological sequences (such as protein or RNA) within a particular family are related by evolution and therefore may be very similar to each other. In this case, applying a standard approach of randomly splitting the data into training and test sets could yield test sequences that are nearly identical to some sequence in the training set, and the resultant benchmark may overstate the model's performance. This motivates the design of strategies for dividing sequence families into dissimilar training and test sets. To this end, we used ideas from computer science involving graph algorithms to design two new methods for splitting sequence data into dissimilar training and test sets. These algorithms can successfully produce dissimilar training and test sets for more protein families than a previous approach, allowing us to include more families in benchmark datasets for biological sequence analysis tasks.

EddyRivasLab/easel/tree/develop. The code used to generate the figures in this paper is available at https://github.com/spetti/split_for_benchmarks.

**Funding:** SP is funded by the NSF-Simons Center for Mathematical and Statistical Analysis of Biology at Harvard (award number #1764269, https://quantbio.harvard.edu/mathbio). SRE is funded by the National Human Genome Research Institute of the National Institutes of Health under Award Number R01-HG009116 (https://www.genome.gov/). The content is solely the responsibility of the authors and does not necessarily represent the official views of the National Institutes of Health. The funders had no role in study design, data collection and analysis, decision to publish, or preparation of the manuscript.

**Competing interests:** The authors have declared that no competing interests exist.

This is a *PLOS Computational Biology* Methods paper.

## Introduction

Computational methods are typically benchmarked on test data that is separate from the data that were used to train the method [1–4]. In many areas of machine learning and statistical inference, data samples can be thought of as approximately independent samples from some unknown distribution describing the data. In this case a standard approach is to randomly split available data into a training and a test set, fit a model to the training set, and evaluate the model on the test set. In computational biology, families of biological sequences are not independent because they are related by evolution. Random splitting typically results in test sequences that are closely related or even identical to training sequence, which leads to artifactual overestimation of performance. The problem becomes more concerning for complex models capable of memorizing their training inputs [5]. This issue motivates strategies that consider sequence similarity and split data into dissimilar training and test sets [1–4].

We are specifically interested in benchmarking methods for remote sequence homology detection. We want to test how well a homology detection method, given a homologous clade of sequences as an input, can detect other homologous sequences in a distant outlying clade. The remote homologs $y$ are not from the same distribution as the known sequences $x$; they are drawn from some different distribution $P(y \mid x, t)$, where $x$ are the known sequences and $t$ accounts for evolutionary distances separating remote homolog $y$ from the known examples $x$ on a phylogenetic tree. We can create artificial cases of this by splitting known sequence families phylogenetically at deep ancestral nodes. The difficulty of detecting remote homologs depends more on the distance to the outlying sequences than on details of the tree topology. Therefore, inferring a complete tree topology is unnecessary; it is sufficient and more relevant to have a clear distance-based rule for establishing training and test set splits that are challengingly dissimilar.

Previous work from our group splits a given sequence family into training and test sets using a single-linkage clustering by pairwise sequence identity at a chosen threshold $p$, such as $p = 25\%$ for protein or $p = 60\%$ for RNA [6, 7]. One cluster becomes the training set, and the remaining clusters are the source of test sequences. This procedure is a fast proxy for building a phylogenetic tree from distances based on percent identity and selecting test sequences from the outlying clades. We refer to this procedure as the Cluster algorithm in this paper. The procedure guarantees that no sequence in the test set has more than $p\%$ pairwise identity to any sequence in the training set. This is a clear and simple rule for ensuring that training and test sets are remotely homologous, and we can control $p$ to vary the difficulty of the benchmark.

We have found that in many cases, the Cluster algorithm is unable to split a family because single-linkage clustering collapses it into a single cluster, but a valid split could have been identified if we removed certain sequences before clustering. For example, if a family contains two groups that would form separate single-linkage clusters at 25% identity and even just one bridging sequence that is >25% identical to a sequence in each group, then single-linkage clustering collapses all the sequences into one cluster. If we omit the bridge sequence, the two groups form separate clusters after single-linkage clustering. The larger the family, the more likely it is to contain sequences that bridge together otherwise dissimilar clusters, so the procedure fails more often on alignments with many sequences. This is a concern because we and

others are exploring increasingly complex and parameter-rich models for remote sequence homology recognition that can require thousands of sequences for training [8–13]. A phylogenetic approach that attempts to identify an out-group would face this same "bridge" issue. In order to produce training/test set splits for benchmarks that cover a more diverse range of sequence families represented by alignments with many sequences, we were interested in improving on Cluster.

Here we describe two improved splitting algorithms called Blue and Cobalt that are derived from "independent set" algorithms in graph theory. A main intuition is that Blue and Cobalt can exclude some sequences as they identify dissimilar clusters. Blue splits more families, but can be computationally prohibitive on alignments with many sequences (over 50,000). Cobalt (a shade of Blue) is much more computationally efficient and is still a large improvement over Cluster. We compare these algorithms to Cluster and to a simple algorithm that selects a training set independently at random, which we call Independent Selection. We compare splitting success and computational time on a large set of different MSAs with 10's to 100,000's of sequences. In addition, we compare homology search benchmarks built with these different splitting algorithms.

## Results

Given a set of sequences (here, a multiple sequence alignment), the goal is to split it into a training set and a test set, such that no test sequence has $> p\%$ pairwise identity to any training sequence and no pair of test sequences is $> q\%$ identical. The first criterion defines dissimilar training and test sets, and the second criterion reduces redundancy in the test set. (We preserve the alignment of the training set, including sequence redundancy; the goal of the benchmark is to have realistic query sequence alignments, which do often include redundant sequences. Different homology search methods deal with sequence redundancy in different ways. Most profile construction methods use relative sequence weights, downweighting similar sequences.) The choice of thresholds $p$ and $q$ should be decided based on the goals of the methods being benchmarked.

In order to guarantee that no test sequence has $> p\%$ pairwise identity to any training sequence, some sequences will end up in neither the training nor the test set. Our algorithms find training and test sets, and they are returned if they are larger than the user-specified minimum acceptable size for each.

We cast the splitting problem in terms of graph theory with each sequence represented by a vertex and similarity indicated by an edge. For example, a pairwise identity of $> p\%$ between two sequences defines an edge for the first criterion. Each splitting method is a two step procedure, for which we use related algorithms. In the first step, we identify disjoint subsets $S$ and $T$ of our original set of sequences, such that for any $x \in S$ and $y \in T$ there is no edge (pairwise identity $> p\%$) between $x$ and $y$. We assign $S$ as the training set and $T$ as the candidate test set. The second step then starts with a graph on $T$, using pairwise identity threshold $q$ to define edges. We identify a representative subset $U$ such that no pair of vertices $y, y' \in U$ is connected by an edge and assign $U$ to be the test set. The graph problems in steps (i) and (ii) are related. It is useful to discuss the simpler algorithm for step (ii) before describing its adaptation to task (i).

Task (ii) is exactly the well-studied graph algorithm problem of finding an independent set in a graph. Formally, in a graph $G = (V, E)$ with vertex set $V$ and edge set $E$, a subset of vertices $U \subseteq V$ is an *independent set* (IS) if for all $u, w \in U$, $(u, w) \notin E$. To frame task (i), we define a *bipartite independent pair* (BIP) as a pair of disjoint sets $U_1, U_2$ such that there are no edges between pairs of vertices in $U_1$ and $U_2$, i.e. for all $u_1 \in U_1$ and $u_2 \in U_2$, $(u_1, u_2) \notin E$. The

algorithms we describe here follow this two-step approach, but differ in how they achieve each step.

While it is NP-hard to find a maximum size independent set in a graph [14], randomized algorithms can be applied to quickly find a maximal independent set (an independent set where if any additional vertex were added, the set would no longer be an independent set). The Blue and Cobalt methods are inspired by two such algorithms [15, 16]. Unlike Cluster, Blue and Cobalt always find a maximal independent set in task (ii).

## Splitting algorithms

In our descriptions below, vertex $w$ is a *neighbor* of vertex $v$ if $(v, w)$ is an edge in the graph. The *degree* of a vertex $v$, denoted $d(v)$, is the number of neighbors of $v$. The *neighborhood* of $v$ in the graph $G = (V, E)$ is $N(v) = \{w \in V : (w, v) \in E\}$.

**Cobalt.** The Cobalt algorithm is an adaptation of the greedy sequential maximal independent set algorithm, studied in [15]. The graph's vertices are ordered arbitrarily, and each vertex is added to the independent set if none of its neighbors have already been added. Step 2 of Cobalt is this algorithm with the vertex order given by a random permutation. Assigning a vertex to an IS disqualifies all of its neighbors from the IS, and so it may be advantageous to avoid placing large degree vertices in the IS. In Cobalt, higher degree vertices are less likely to be added to the IS; a vertex $v$ is placed in the IS if all of its neighbors come after it in the random order, which happens with probability $1/d(v)$. The bias towards including low degree vertices, which correspond to "outlier" sequences, is desirable for creating benchmarks that include the most remote homologs.

```
Algorithm 1: Greedy sequential IS in graph G = (V, E) (Cobalt Step 2)
Result: An independent set U in G = (V, E)
U = ∅
Place the vertices of V in a random order: v₁, v₂, ... vₙ.
for i=1 to n do
  if vᵢ is not adjacent to any vertex in U then U = U ∪ {vᵢ};
end
return U
```

Step 1 is a variant which instead finds a bipartite independent pair. Once a BIP is found in Step 1, the larger set is declared the training set, and the smaller set is input into the greedy sequential IS algorithm as the vertex set of $G_2$ (Cobalt Step 2). We assign the larger set as the training set because the goal is to benchmark on realistic input alignments, and the larger cluster is more like the original input alignment; additionally, we aim to benchmark methods that may require large numbers of training sequences.

```
Algorithm 2: Greedy sequential BIP in graph G = (V, E) (Cobalt Step 1)
Result: A bipartite independent pair S, T in G = (V, E)
S, T = ∅
Place the vertices of V in a random order: v₁, v₂, ... vₙ.
for i=1 to n do
  Sample r ~ unif(0, 1).
  if r < 1/2 then
    if vᵢ is not adjacent to any vertex in S then S = S ∪ {vᵢ};
    else if vᵢ is not adjacent to any vertex in T then T = T ∪ {vᵢ};
  else
    if vᵢ is not adjacent to any vertex in T then T = T ∪ {vᵢ};
    else if vᵢ is not adjacent to any vertex in S then S = S ∪ {vᵢ};
  end
end
if |S| < |T| then swap the names of S and T;
return S, T
```

**Blue.** The Blue algorithm leverages the fact that the number of vertices disqualified by the addition of a vertex $v$ to an IS is not exactly its degree; it is the number of neighbors of $v$ that are still eligible. Blue is based on the IS Random Priority Algorithm introduced by [16]. In each round of this algorithm, the probability of selecting a vertex is inversely proportional to the number of neighbors that are eligible at the beginning of the round.

Each eligible vertex is labeled with a value drawn uniformly at random from the interval $[0, 1]$. If a vertex has a lower label than all of its neighbors, the vertex is added to the independent set and its neighbors are declared ineligible. This process repeats until there are no eligible vertices. The pseudocode presented here describes the multi-round election process in the most intuitive way. Our implementation avoids storing the entire graph structure $G$ and instead only computes the similarity relationship when algorithm needs to know whether an edge exists.

```
Algorithm 3: Random Priority IS in graph G = (V, E) (Blue Step 2)
Result: An independent set U in G = (V, E)
U = ∅; L = V
while L ≠ ∅ do
  Declare ℓ an empty dictionary.
  for each v ∈ L do ℓ(v) ∼ unif(0, 1);
  Place the vertices of L in a random order: v₁, v₂, ... vₖ
  for i=1 to k do
    if vᵢ ∈ L and ℓ(vᵢ) < ℓ(w) for all w ∈ L ∩ N(vᵢ) then
      U = U ∪ {vᵢ}
      L = L \ (N(vᵢ) ∪ {vᵢ})
    end
  end
end
return U
```

In our modification of this algorithm to find a BIP, we keep track of each vertex's eligibility for each of the sets $S$ and $T$. In each round, every vertex that is eligible for at least one set is declared either an $S$-candidate or $T$-candidate and assigned a value uniformly at random from the interval $[0, 1]$. Each $S$-candidate is added to $S$ if its label is smaller than the labels of all its neighbors that are both $T$-candidates and $T$-eligible. When a vertex $v$ is added to $S$, $v$ is declared ineligible for both $S$ and $T$, and all neighbors of $v$ are declared ineligible for $T$. After iterating through all $S$-candidates, any $T$-candidates that are still $T$-eligible are added to $T$. Once a BIP is found, the larger set is declared the training set, and the smaller set is input into the greedy sequential IS algorithm as the vertex set of $G_2$ (Blue Step 2).

```
Algorithm 4: Random Priority BIS in graph G = (V, E) (Blue Step 1)
Result: A bipartite independent pair S, T in G = (V, E)
S, T = ∅; L_S, L_T = V
while L_S ∪ L_T ≠ ∅ do
  C_S, C_T = ∅
  for each v ∈ L_S ∪ L_T do
    if v ∈ L_S \ L_T then C_S = C_S ∪ {v};
    if v ∈ L_T \ L_S then C_T = C_T ∪ {v};
    if v ∈ L_T ∩ L_S then
      Sample r ∼ unif(0, 1).
      if r < 1/2 then C_S = C_S ∪ {v};
      else C_T = C_T ∪ {v};
    end
  end
  Declare ℓ an empty dictionary.
  for each v ∈ C_S ∪ C_T do ℓ(v) ∼ unif(0, 1);
  Place the vertices of C_S in a random order: v₁, v₂, ... vₖ
  for i=1 to k do
```

```
          if ℓ(vᵢ) < ℓ(w) for all w ∈ L_T ∩ C_T ∩ N(vᵢ) then
            S = S ∪ {vᵢ}, L_T = L_T \ (N(vᵢ) ∪ {vᵢ}) and L_S = L_S \ {vᵢ}
          end
        end
        T = T ∪ (C_T ∩ L_T)
        for v ∈ (C_T ∩ L_T) do L_T = L_T \ {v} and L_S \ (N(v) ∪ {v});
      end
    if |S| < |T| then swap the names of S and T;
    return S, T
```

**Repetitions of Blue and Cobalt.** The use of randomness is a strength of Cobalt and Blue. Unlike Cluster, which produces the same training set and same test set size every time the algorithm is run, the sets produced by Blue and Cobalt may be highly influenced by which vertices are selected first. Running the algorithms many times typically yields different results. We implemented two features to take advantage of this: (i) the "run-until-$n$" option in which the algorithm runs at most $n$ times and returns the first split that satisfies a user defined threshold, and (ii) the "best-of-$n$" option in which the algorithm runs $n$ times and returns the split that maximizes the product of the training and test set sizes (i.e., the geometric mean).

**Cluster.** In the first step, the graph $G_1$ is partitioned into connected components; by definition there is no edge between any pair of connected components. The vertices of the largest connected component are returned as the training set $S$. The remaining vertices become the set $T$, and the training set $U$ is formed by selecting one vertex at random from each connected component of the graph $G_2$ with vertex set $T$.

**Independent selection.** In the first step, every vertex of $G_1$ is added to set $S$ independently with probability $p = 0.70$. All vertices that are not in $S$ and not adjacent to any vertex in $S$ are added to $T$. In the second step, the Greedy sequential IS algorithm (Cobalt Step 2) is applied to $G_2$ (which has vertex set $T$) to produce a training set $U$.

## Performance comparisons

We compared the success rates for splitting biological sequence families of different sizes by running our algorithms on multiple sequence alignments from the protein database Pfam [17]. To study a wide range of different numbers of sequences per family, we split both the smaller curated Pfam "seed" alignments and the larger automated "full" alignments.

Fig 1 illustrates the pass rates of the algorithms when $p = 25\%$ and $q = 50\%$. By setting $p = 25\%$, we are testing how well each homology search method can identify previously unseen distant homologs that are at most 25% identical to a training sequence. Of the 12340 Pfam seed families with at least 12 sequences, Blue splits 34.4%, Cobalt splits 29.0%, Cluster splits 19.1%, and Independent Selection splits 6.8% into a training-test set pair with at least 10 training and 2 test sequences. After running Blue and Cobalt 40 times each, 59.8% and 55.9% of the families (respectively) are successfully split. For the Pfam full families, we require that the training and test sets have size at least 400 and 20 respectively. Of the 9827 Pfam full families with at least 420 sequences, Blue splits 30.5%, Cobalt splits 28.4%, Cluster splits 14.0%, and Independent Selection 3.0%. The algorithms were considered unsuccessful on the 188, 2, and 1 families that Blue, Cluster, and Cobalt did not finish in under 24 hours. The success rates of Blue and Cobalt increase to 53.6% and 50.1% after 40 iterations.

Fig 2 illustrates the characteristics of the full families that are successfully split by the algorithms at the 400/20 threshold. S1 Fig is the analogous plot for the seed families at the 10/2 threshold. The algorithms struggle to split smaller families and families in which a high fraction of the sequence pairs are at least 25 percent identical. S2 and S3 Figs illustrate the sizes of the training and test sets produced by the four algorithms.

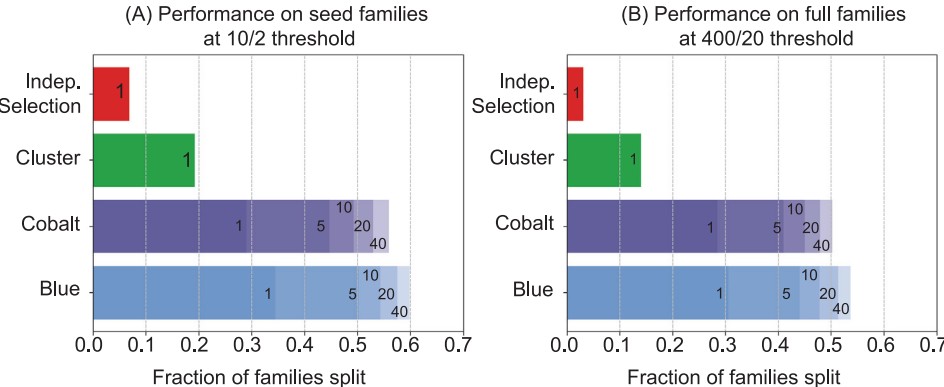

**Fig 1. Performance of splitting algorithms on Pfam families.** (A) Fraction of the 12340 Pfam seed families with at least 12 sequences that were split into a training set of size at least 10 and test set of size at least 2. The numbers on the Blue and Cobalt bars indicate the fraction of families successfully split at least once out of 1, 5, 10, 20, 40 independent runs. (B) Fraction of the 9827 Pfam families with at least 420 sequences in their full alignment that were split into a training set of size at least 400 and test set of size at least 20.

We also compare the running times of our implementations of each algorithm. Table 1 displays the runtime of the algorithms on the multi-MSAs for the Pfam seed and full databases. All algorithms can split the entire Pfam seed database in under four minutes. Most Pfam full families can be split in under one minute. Fig 3 illustrates the runtimes as a function of the product of the number of sequences and the columns in the alignment. Our implementations take as input a set of $N$ sequences and only compute the distance between a pair of sequences if the algorithm needs to know whether there is an edge between the corresponding vertices. In the worst case (a family with no edges), our algorithm must compute $O(N^2)$ distances. Computing percent identity is $O(L)$ where $L$ is the length of the sequence. Therefore when distance is percent identity, the worst case runtime is $O(LN^2)$.

## Benchmarking homology search methods with various splitting algorithms

All four algorithms produce splits that satisfy the same dissimilarity criteria ($p = 25\%$ and $q = 50\%$), but we noticed that the different procedures create training-test set pairs that are more or less challenging benchmarks. To study this, we used the four algorithms in a previously published benchmark procedure [7] that evaluates a method's ability to detect whether a sequence contains a subsequence homologous to a Pfam family. Briefly, negative decoy sequences are synthetic sequences generated from shuffled subsequences randomly selected from UniProt, and positive sequences are constructed by embedding a single test domain sequence into a synthetic sequence.

We applied each algorithm to the Pfam seed families with the requirement that there be at least 10 training and 2 test sequences. To avoid over-representing families that yielded large test sets, all test sets were down-sampled to contain at most 10 sequences. First we used these splits to benchmark profile searches with the HMMER hmmsearch program [18]. As illustrated by Fig 4, ROC curves vary substantially based on the splitting algorithm used. The reported accuracy for hmmsearch is highest for the benchmark produced by Independent Selection, followed by the benchmarks produced by Cobalt, Blue, and then Cluster.

We consider two hypotheses for why HMMER performance depends on the splitting method: (i) the families that are successfully split by a particular algorithm are also inherently easier or harder for homology recognition, and (ii) the splitting algorithms create training and test sets with inherently different levels of difficulty.

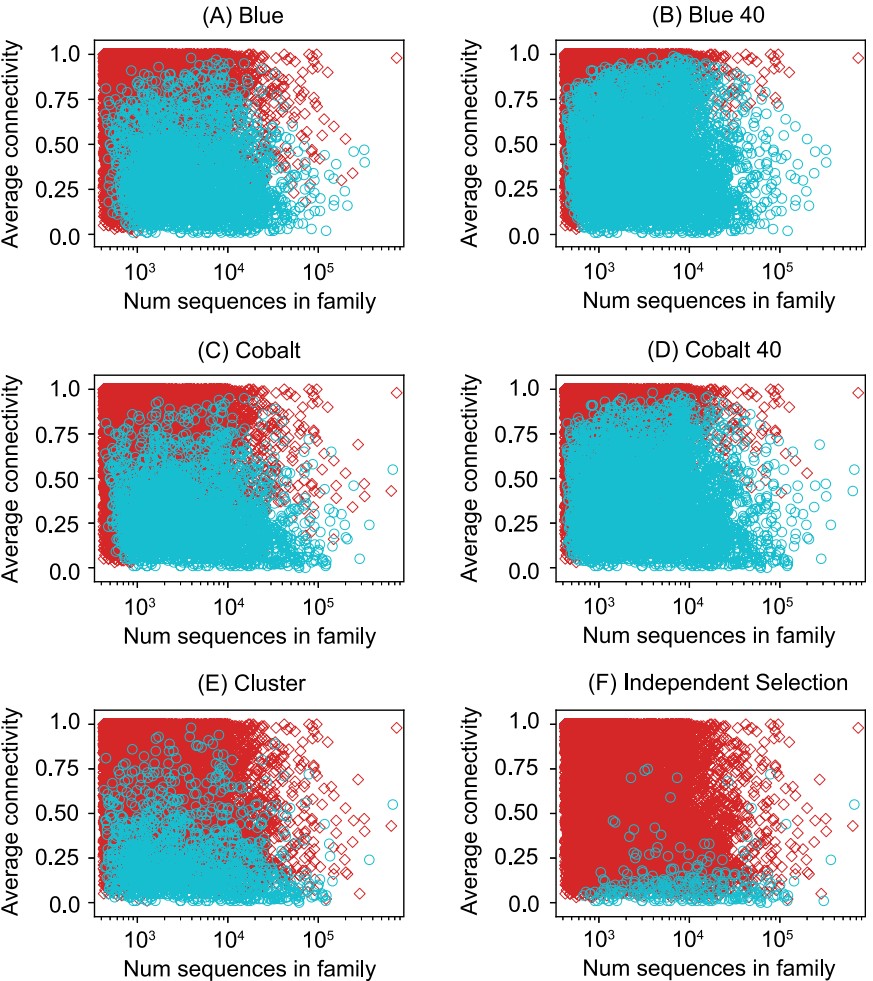

**Fig 2. Characteristics of Pfam full families successfully split.** Each marker represents a family in Pfam. The connectivity of a sequence is the fraction of other sequences in the full family with at least 25% pairwise identity. Families successfully split into a training set of size at least 400 and a test set of size at least 20 are marked by a cyan circle, whereas families that were not split are marked by a red diamond. In (B) and (D) the cyan circle represents at least one successful split among 40 independent runs. The 34 families that Blue did not finish splitting within 6 days are not included in the Blue plots.

To explore the first hypothesis, we compiled ROC curves for the 708 families split by all four algorithms. Fig 4B shows that the ROC curves for Blue and Cobalt are brought closer to the ROC curve for Independent Selection, and so hypothesis (i) may explain some of the discrepancy between the Blue, Cobalt, and Independent Selection benchmarks. However,

**Table 1. Runtime of implementations on Pfam seed and full.** The runtime benchmarks were obtained by running each algorithm on the seed and full multi-MSAs Pfam-A.seed and Pfam-A.full on 2 cores with 8 GB RAM for the seed alignments and on 3 cores with 12 GB RAM for the full alignments. We did not compute the maximum runtime of the Blue algorithm; the algorithm failed to terminate within 6 days for 34 families.

| Algorithm | All seed (min:sec) | All full (days-hours:min) | Max full (hours:min) | Full families >1 min |
|---|---|---|---|---|
| Blue | 3:16 | — | — | 1422 (7.9%) |
| Cobalt | 0:43 | 7–0:24 | 46:25 | 419 (2.3%) |
| Cluster | 0:58 | 5–0:31 | 37:17 | 244 (1.3%) |
| Indep. Selection | 0:19 | 0–5:49 | 1:30 | 48 (0.2%) |

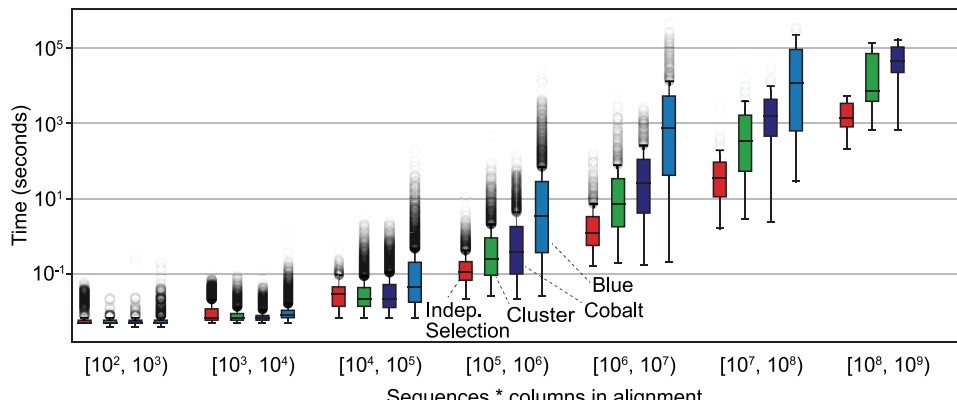

**Fig 3. Runtime of algorithms.** Each algorithm was run once on each Pfam seed and full alignment for at most 6 days. The runtimes are reported as a function of the product of the number of sequences and the number of columns in the alignment, as bar plots including outliers (translucent grey circles). The boxes extend from the first to third quartile, and the median is marked by a horizontal line. The results for families with at most 10,000 sequences were obtained on 2 cores and 8 GB of RAM, and the remaining were obtained on 3 cores and 12GB of RAM. The results do not include 34 families that Blue did not finish running within 6 days. Blue finished 939 of 944 families in the $[10^6, 10^7)$ range, 58 of 85 families in the $[10^7, 10^8)$ range, and 1 of 3 families in the $[10^8, 10^9)$ range (and we omitted a bar plot for Blue for $[10^8, 10^9)$).

hypothesis (i) does not explain the discrepancy with the Cluster benchmark because the Blue and Cobalt ROC curves are even farther from the Cluster ROC curve under the family restriction.

The second hypothesis is likely a better explanation. A sequence that is less than 25% identical to all other sequences in the family is probably the hardest sequence for a homology search program to recognize. If such a sequence exists, the Cluster algorithm will always assign it to the test set, whereas Blue, Cobalt, and Independent selection will assign it to the test set 50, 50, and 30 percent of the time respectively. Fig 4C illustrates distribution of distances (in percent identity) between each sequence in the test set and the closest sequence in the training set. The test sequences are on average farther from the closest training sequence under the Cluster algorithm.

Since the different algorithms lead to different performance results with one homology search program, we then wanted to see if the choice of splitting algorithm alters the relative performance in a comparison of different homology search algorithms. Fig 5 demonstrates that the relative ranking of the performance of various homology search algorithms is approximately the same regardless of which splitting algorithm was used to produce the split of the data into training and test sets. In addition to HMMER, we benchmarked BLASTP, PSI-BLAST, and DIAMOND. PSI-BLAST performs a BLAST search with a position-specific scoring matrix determined in our case from the set of training sequences [19]. DIAMOND is a variant BLASTP that utilizes double indexing, a reduced alphabet, and spaced seeds to produce a faster algorithm [20]. DIAMOND is benchmarked using "family pairwise search," in which the best E-value between the target sequence (positive test or negative decoy) and all sequences in the training set is reported [21]. DIAMOND is designed for speed, not sensitivity, and its low sensitivity is apparent. Running DIAMOND with the "sensitive" flag (denoted diamond-sen in Fig 5) improves accuracy, but it remains less accurate than PSI-BLAST, BLASTP, and HMMER. The choice of splitting algorithm does not alter the relative order of performance of the four search algorithms.

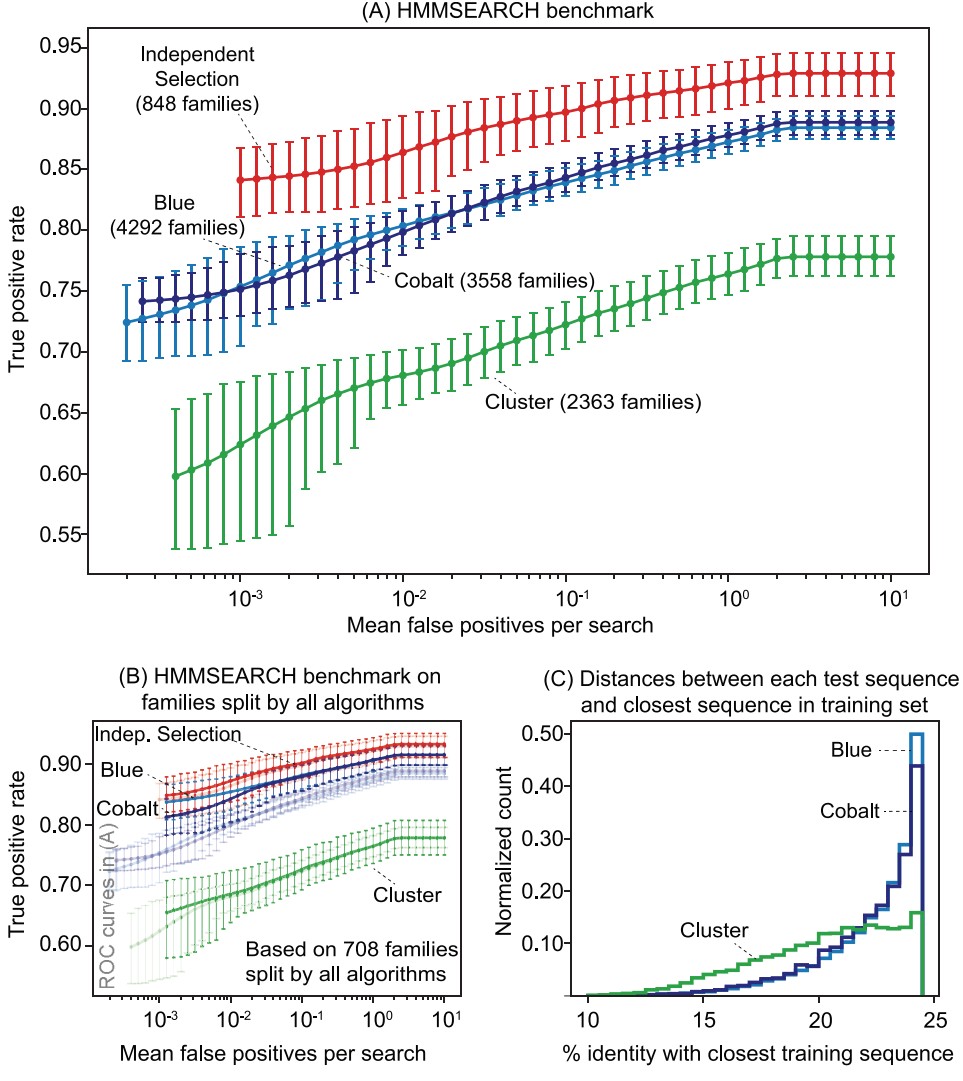

**Fig 4. Benchmarks of HMMSEARCH.** (A) Each benchmark includes data from all families that were split into training and test sets of size at least 10 and 2 respectively by one run of the algorithm. The number of families included in the benchmark for each algorithm is stated in the labels. For each family, HMMER produces a single profile from the alignment of the training sequences. We constructed 200,000 decoy sequences from shuffled subsequences chosen randomly from UniProt. At most 10 positive test sequences are constructed by embedding a single homologous domain sequence from the test set into synthetic decoy sequence. (See Methods) The $x$-axis represents the number of false positives per profile search and the $y$-axis represents the fraction of true positives detected with the corresponding E-value, over all profile searches. The error bars at each point represent a 95 percent confidence interval obtained by a Bayesian bootstrap. (B) The faded lines are copies of the plot (A). The dark lines are the analogous curves constructed by restricting to the benchmarks to the 708 families successfully split by all four algorithms. (C) The distribution of the distances between each test sequence and the closest training sequence (measured in percent identity) for families split by Blue, Cobalt, and Cluster.

## Discussion

We present two new algorithms, Blue and Cobalt, that are able to split more Pfam protein sequence families into training and test sets so that no training-test sequence pair is more than $p = 25$ percent identical and no test-test sequence pair is more than $q = 50$ percent identical. Our algorithms are able to split approximately three times as many Pfam families as compared to the Cluster algorithm we have used in previous work [6, 7, 10], and more than six times as

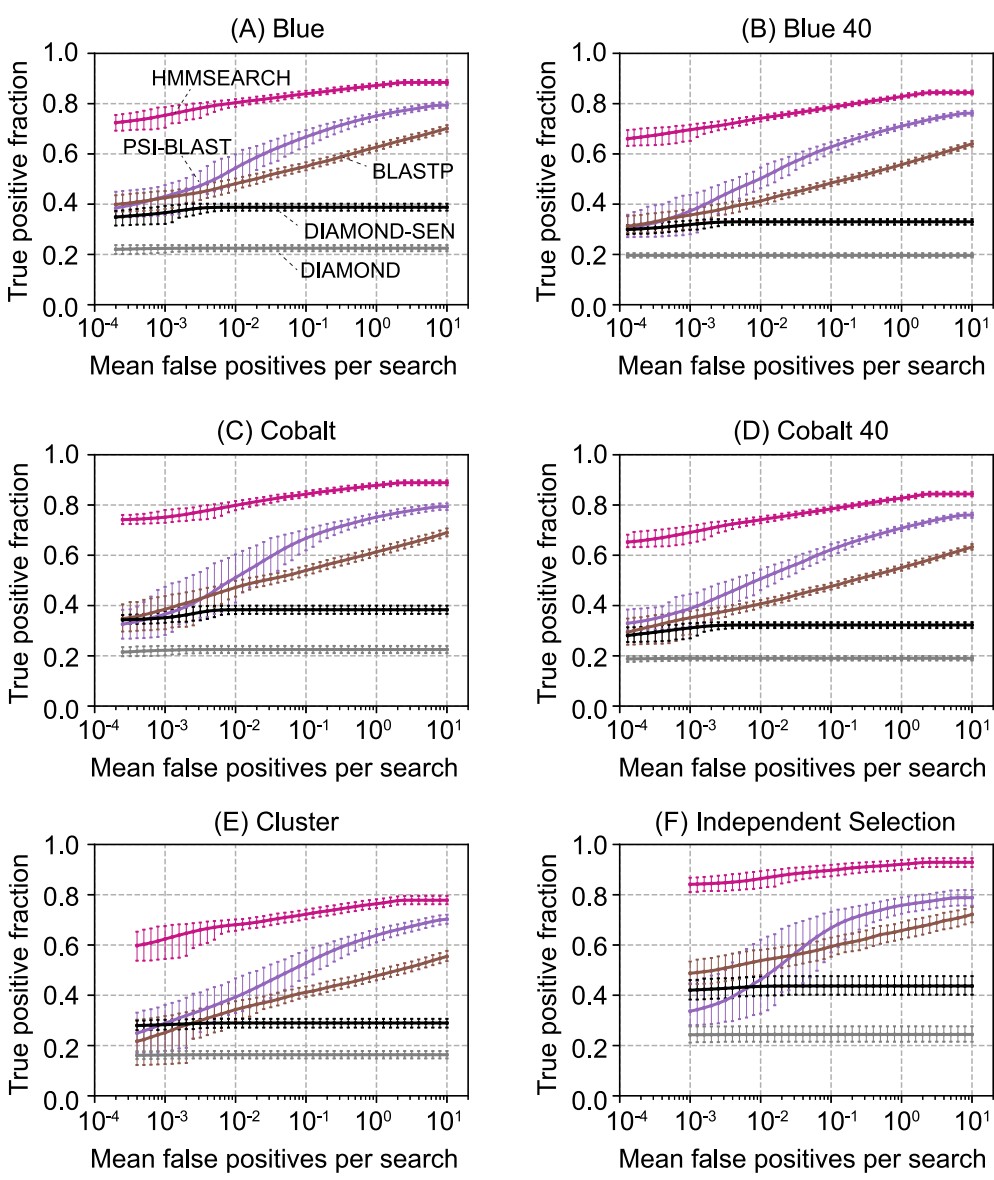

**Fig 5. Homology search benchmarks on data produced by splitting algorithms.** The benchmarks are constructed as in Fig 4. Blue 40 and Cobalt 40 refer to the algorithms run with the "best-of-40" feature. BLASTP and DIAMOND are benchmarked using family pairwise search.

many families as compared to a simple Independent Selection algorithm (see Fig 1). Our algorithms allow us to create larger and more diverse benchmarks across more Pfam families, and also to produce training sets with thousands of sequences for benchmarks of new parameter-rich machine learning models. The Blue algorithm maximizes the number of families included; the faster Cobalt algorithm is recommended for splitting large sequence families.

Blue and Cobalt are random algorithms that typically create different splits each time they are run. Although this is useful, different splits are unlikely to be independent. The variation between splits will depend on the structure of the graph for the sequence family. Different splits are not suited for a procedure like *k*-fold cross-validation in machine learning, for example.

We were initially surprised to find that for the same sequence identity thresholds, the four splitting algorithms result in benchmarks of varying challenge levels for homology search algorithms. However, within a given benchmark, relative performance of different algorithms is unaffected by the choice of splitting algorithm. Moreover, since the dissimilarity requirement $p$ is an input, the difficulty of a benchmark is tunable.

These algorithms address a fundamental challenge in training and testing models in biological sequence analysis. Random splitting into training and test data assumes that all data points are independently and identically drawn from an unknown distribution $P(x)$. A model of $P(x)$ is fitted to the training data and evaluated on the held-out test data. In contrast, the remote homologs $y$ that we are interested in identifying come from a different distribution than the the known sequences $x$. The distribution $P(y \mid x, t)$ depends on both the known sequences $x$ and some measure of the evolutionary distance between the homologs $t$. In machine learning, "out of distribution" recognition typically means flagging anomalous samples, but this is a case where it is the task itself [22]. Our procedures create out-of-distribution test sets, with dissimilarity of the training/test distributions controlled by the pairwise identity parameter $p$. The out-of-distribution nature of the remote homology search problem affects not only how appropriate benchmarks are constructed, but also how improved methods are designed.

## Materials and methods

### Details of benchmarking procedure

We used the benchmarking pipeline as described in [7], as implemented in the "profmark" directory and programs in the HMMER software distribution. Briefly: for a given input multiple sequence alignment (MSA), first remove all sequences whose length is less than 70% of the mean. Then the splitting algorithm produces a training set and a test set. The training set sequences remain aligned according to the original MSA, and the sequence order is randomly permuted. This alignment is used to build a profile in benchmarks of profile search methods such as HMMER "hmmsearch" and PSI-BLAST.

The test set is randomly down-sampled to contain at most 10 sequences. Pfam MSAs consist of individual domains, not complete protein sequences. Each test domain sequence is embedded in a synthetic nonhomologous protein sequence as follows: (i) draw a sequence length from the distribution of sequence lengths in UniProt that is at least as long as the test domain (ii) embed the test domain at a random position, (iii) fill in the remaining two segments with nonhomologous sequence by choosing a subsequence of the desired length from UniProt and shuffling it. The resultant sequences form the positive test set for the particular family. Next form a shared negative test set of 200,000 sequences similarly as follows: (i) choose a positive test sequence at random (from the full group of test sequences) and record the lengths of the three segments, (iii) fill in each segment as described in step (iii) of producing positive sequences. The default "profmark" procedure in HMMER embeds two test domains per positive sequence (for purposes of testing multidomain protein parsing); for this work we used the option of embedding one domain per positive sequence.

### Hardware, software and database versions used

All computations were run on Intel Xeon 6138 Processors at 2.0 Ghz. Our time benchmarks were measured in real (wall clock) time. Our tests were performed on the Pfam-A 33.1 database, released in May 2020. We used UniProt release 2/2019. Software versions used: HMMER 3.3.1, BLAST+ 2.9.0, DIAMOND 0.9.5.

## Supporting information

**S1 Fig. Characteristics of Pfam seed families successfully split.** Each marker represents a family in Pfam. The connectivity of a sequence is the fraction of other sequences in the seed family with at least 25% pairwise identity. Families successfully split into a training set of size at least 10 and a test set of size at least 2 are marked by a cyan circle, whereas families that were not split are marked by a red diamond. In (B) and (D) the cyan circle represents at least one successful split among 40 independent runs.
(TIF)

**S2 Fig. Size of training and test sets produced by each algorithm on seed families.** The two-dimensional normalized histograms illustrate the distribution of training and test set sizes produced by the algorithms among results with at least 10 and 2 training and test sequences respectively. In each plot, the $x$-coordinate and $y$-coordinates of the green circle represent the median training and median test set sizes respectively. The white X is placed at the median training and test set sizes among the 2363 families that were successfully split by Blue, Cobalt, and Cluster.
(TIF)

**S3 Fig. Size of training and test sets produced by each algorithm on full families.** The two-dimensional normalized histograms illustrate the distribution of training and test set sizes produced by the algorithms among results with at least 400 and 20 training and test sequences respectively. In each plot, the $x$-coordinate and $y$-coordinates of the green circle represent the median training and median test set sizes respectively. The white X is placed at the median training and test set sizes among the 1070 families that were successfully split by Blue, Cobalt, and Cluster.
(EPS)

## Acknowledgments

The computations in this paper were run on the Cannon cluster supported by the FAS Division of Science, Research Computing Group at Harvard University.

## Author Contributions

**Conceptualization:** Samantha Petti, Sean R. Eddy.

**Data curation:** Samantha Petti.

**Formal analysis:** Samantha Petti.

**Funding acquisition:** Samantha Petti, Sean R. Eddy.

**Methodology:** Samantha Petti.

**Resources:** Sean R. Eddy.

**Software:** Samantha Petti.

**Supervision:** Sean R. Eddy.

**Visualization:** Samantha Petti.

**Writing – original draft:** Samantha Petti.

**Writing – review & editing:** Samantha Petti, Sean R. Eddy.

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
