## [Decision Letter · Decision Letter 0]

4 Nov 2021

Dear Dr. Petti,

Thank you very much for submitting your manuscript "Constructing benchmark test sets for biological sequence analysis using independent set algorithms" for consideration at PLOS Computational Biology.

As with all papers reviewed by the journal, your manuscript was reviewed by members of the editorial board and by several independent reviewers. In light of the reviews (below this email), we would like to invite the resubmission of a significantly-revised version that takes into account the reviewers' comments. In particular, we encourage you to address better explain the meaning of "independent" data and justify the use of percent of Identity (PID) as opposed to an estimated phylogeny to define the split into training and test data.

We cannot make any decision about publication until we have seen the revised manuscript and your response to the reviewers' comments. Your revised manuscript is also likely to be sent to reviewers for further evaluation.

Sincerely,

Maricel G Kann

Associate Editor

PLOS Computational Biology

Feilim Mac Gabhann

Editor-in-Chief

PLOS Computational Biology

Reviewer's Responses to Questions

**Comments to the Authors:**

Reviewer #1: This article is a model of clarity in presentation, and I have very few optional suggestions for improvement.

p3, line 71. You may have views about reducing redundancy in the training set, too. It might be interesting to state them.

p5, line 101. The present article presents a complete and unified investigation. In several of your algorithms, however, one could retain some randomness after ordering the candidate lists not by random permutation, but by degree class (small to large) and then permuting randomly within each class of degrees. The effect would be to prioritize vertices of lower degree in the test and training sets, i.e., to ensure inclusion of outliers wherever possible. I suggest future examination of this variant of your algorithms.

p.9, line 135. The article makes a good case for the virtues of randomness, so the previous comment is intended entirely as a suggestion.

p.10, Figure 1. The X-axis is unlabeled. The legend states that it is "Fraction of families successfully split", but adding the axis label would help the top label "Performance..." (which can be retained). I admire the authors' ingenuity in introducing several dimensions of information into the plot.

p.14, Figure 4. PID = percent identity. I could not find the full acronym, which should be explicit.

p.14, line 214. It would be interesting to know if the ordering by degree changed the difficulty of the search.

p.18, line 257. Who benchmarks the benchmarks? The observation that the benchmark changes the difficulty of the test search (and possibly, therefore, the robustness of training) but preserves the relative performance of the algorithms is very reassuring.

Reviewer #2: This study present three new methods, INDEPENDENT SELECTION, BLUE, and COBALT, that enable an input set of sequences (provided in a multiple sequence alignment) to be split into three sets: the training set, the test set, and the discarded set, so that the pairwise identity (PID) between any sequence in the training set and any sequence in the test set is below a user-provided threshold, and the PID between any two test sequences are below another user-provided threshold. The motivation for this approach is the idea that the training data and the test data need to be independent of each other, which the authors note is difficult due to evolution (i.e., all sequences are related to each other because of sharing a common ancestor). These new methods are designed to improve on an earlier algorithm for the same problem, called "CLUSTER", which also aims to achieve this, but these new approaches employ randomness to produce solutions to this problem that might give better downstream results.

There are several parts to the evaluation. First, they examine how many PFAM families are successfully split to produce a training set with at least 10 sequences and a test set with at least 2 sequences. The second evaluation examines the consequences of using these splits of PFAM families for a specific problem (detecting whether a sequence is locally homologous to a given family). Finally, they also evaluate running time. Putting running time aside, they note that BLUE and COBALT are strictly better than CLUSTER with respect to the first two tests (how many families are split and how accurate the subsequent bioinformatics test is using this split). They also note that INDEPENDENT SELECTION is more accurate than BLUE and COBALT, but since INDEPENDENT SELECTION is successful at splitting a much smaller number of families, this is not that important. The running time comparison shows BLUE is more computationally expensive than the remaining methods.

All this is fine. The description of the methods they developed is easy to follow and natural (i.e., they are using straightforward ideas). The writing is good (though there are a few places that could be improved, given below). The results support the claims, generally.

However, I have two questions about the whole approach that I think would merit some discussion. The first question has to do with whether using PID to define the test and training data really gets at "independence", which the authors have justified by noting that two sequences datasets can be non-independent due to shared evolutionary history. Therefore, if the test data form a small clade in the evolutionary tree, separated by a fairly long branch from the training data, they would have perhaps good separation in terms of PID but would not be independent. Wouldn't a better approach be to construct a phylogeny based on the alignment, and then extract two clades from the phylogeny?

The second question has to do with other aspects of defining training data and test data. An important consideration for training data is that it should be representative of the test data. It seems to me that by enforcing a large PID separation between test and training data in a sense is leading to the training data being very different from the test data. This may result in machine learning methods not generalizing well to the test data, since the test data are so different. This hypothesis is consistent with the results shown in the study, since their new methods end up producing splits into test and training data that have *smaller* gaps (in terms of PID) than the original CLUSTER method, and this is used to explain why the new methods have better accuracy. Wouldn't this also potentially suggest that relaxing the required gap between testing and training data even more might further improve accuracy in the subsequent bioinformatics analyses?

In general, therefore, the main questions to the authors are: (1) Is PID a very good proxy for evolutionary independence, and why not instead use an estimated phylogeny to define the split into training and test data? (2) Can the authors discuss the competing objectives in defining the training and test datasets, and characterize the different approaches in terms of these competing objectives?

Finally, there are a few places where the writing could be improved. Some of them are really minor (low-level writing issues), but others have to do with clarity.

1. What are "deep alignments"? (See lines 54 and 59)

2. The problem formulation (beginning of "Results") clearly depends on parameters p and q. Their selection should be discussed, and the problem formulation should note this dependency. Also, since they will remove sequences, the problem formulation is inadequate: they need to note something about removing sequences (perhaps not too many). Then also perhaps something about choosing between alternative splittings of the input: what are the criteria?

3. The choice of what set becomes the training set and which one becomes the test set is not really discussed, beyond saying (lines 34 and 35) "one cluster (usually the largest one) becomes the training set". Is that wise? Isn't there a possibility that the larger one might be the easier or the harder one? Wouldn't picking the set that has the largest amount of heterogeneity provide a better training set?

4. Lines 189-195. This is where the evaluation process is described. It would merit an additional sentence, e.g., "Here we test the ability of HMMER to detect whether a sequence contains a substring that is homologous to a family in PFAM" or perhaps something else? That way the "true negatives" and "true positives" are understood.

Low-level Writing:

• Line 67. "Given set" -> "Given a set".

• Line 143. Unless British English conventions are used, add a comma after "i.e."

• Lines 144-145. By definition, a connected component is a maximal subset that is connected, therefore by definition there cannot be any edge between two connected components. This sentence should be rephrased. For example, replace "partitioned into connected components, such that" by "partitioned into connected components; by definition,"

Reviewer #3: This paper is extremely clear and well-written (mostly, see below).

It concerns getting subsets of sequences that are not too closely

related to each other. This may be of great practical importance for

learning models from training data, and for benchmark assessments.

My main comment is that the motivation, and meaning of "independent"

data, needs better explanation. The usual idea of machine learning is

that same-class objects are nearby in some high-dimensional space, for

some definition of nearby. Does that make them non-independent?

For example, a previous paper ("Benchmarking the next generation of

homology inference tools" Saripella et al. 2016) constructed a

benchmark "aiming to ensure... homologies at different evolutionary

distance". Perhaps that is more representative/realistic, than

ensuring distant homologies only?

In the present paper, the final Discussion paragraph states that

homology search is an "out of distribution" task, but that is not

obviously true. For example, if we learn from mammal sequences then

try to find bacterial sequences, that is indeed out-of-distribution,

but why can't we learn from diverse sequences and thereby be "within

distribution"?

# More minor points:

"Out of distribution" sounds like "transfer learning": may or may not

be worth mentioning.

The "Cluster" method uses one (largest) connected component as the

training set. This seems obviously pessimal, because it tends to

minimize the diversity of the training set. Could this be why the

Cluster benchmark is harder?

Not critical, but it would be interesting to see the runtime breakdown

into the two steps.

The homology benchmark has a severe and easily-fixable problem: the

negative sequences are shuffled thus lack low-complexity/simple

repeats, which are a significant issue in practice. Easy fix: use

reversed instead of shuffled sequences. I realize this is somewhat

tangential to the paper's main point.

It would be helpful to briefly summarize what is known about the

"well-studied" graph problem of independent sets. Maybe mention

maximal versus maximum. Finding a maximum set (or even approximating

it?) is theoretically hard. Are randomized algorithms known to be

advantageous?

Reviewer #4: Description: This paper addresses an important but often ignored problem with protein training and test sets, namely that, when sequences are randomly sampled to each set, the training set often contains nearly the same sequences as the test set. Consequently, protein domain models that are trained on the training set may appear to perform better on the test set than models that would be trained on sequences drawn independently from the underlying distribution. The authors previously addressed this concern using a clustering approach, termed the Cluster algorithm in their paper, that splits the input sequences into test and training sets that share less than a specified percentage of sequence identity. However, they found that this algorithm often fails to split a family due to ‘bridging sequences’ present in the input set. To improve upon current methods, in this study the authors develop two new algorithms, termed Blue and Cobalt, that are derived from “independent set” algorithms in graph theory. The Blue and Cobalt algorithms are better able to split the input set into test and training sets by identifying dissimilar clusters. Blue successfully splits more families than Cobalt, which, however, is computationally more efficient. The authors demonstrate the advantage of these new algorithms by applying them to a large number of Pfam alignments.

Comments: The Blue and Cobalt programs appear to be useful tools for benchmarking search programs and, in particular, are an improvement over the Cluster algorithm, as the authors illustrate using Pfam MSAs. However, this study raises some fundamental questions regarding training and test sets that should at least be discussed.

Comment 1: If an input set is split into two dissimilar sets, it seems likely that these sets will correspond to distinct functionally divergent subgroups. If so, then a protein model obtained using the training set will fail to correspond to the test set: i.e., one will be comparing apples and oranges. In that case a search program might be justified in failing to detect a test sequence given that it belongs to a different subgroup. Of course, what the authors appear to want is a model of the superfamily features upon which one could also superimpose a mixture of the family and subfamily models within that superfamily. Hence, to detect a distant member of the superfamily that corresponds to a subgroup or an orphan sequence absent from the training set, one should perhaps only model the superfamily features. This is often done by purging closely related sequences from the input set so as to retain only the most highly conserved residue positions as well as the more subtle residue propensities at other positions corresponding to the characteristics and the structural core of the superfamily. Alternatively, one could retain all sequences, but down weight them for sequence redundancy rather than purging the set.

Comment 2: Although purging the input set of all but the most diverse sequences seems like a good way to avoid training biases, this can lead to undesirable side effects. For example, input sets often include pseudogene products and sequences corresponding to DNA open reading frames containing frame shifts or splicing artifacts. Since such sequences will lack similarity to functional sequences, they will be overrepresented in the purged set. Hence, focusing too much on the most diverse sequences can, of course, add a significant amount of misleading noise to the training set. Hence, to better capture the key features of both the superfamily and of each subgroup within the superfamily it might make more sense to first group sequences into closely-related clusters (each, to some degree, corresponding to a functionally divergent subgroup) and then select, as a representative for each cluster, that sequence sharing the greatest similarity, on average, with the other sequences in the cluster.

Comment 3: Some protein families are very highly conserved across distantly related organisms. For example, histone H4 from pea and cow differ at only 2 out of the 102 positions. This makes it easy, of course, to distinguish these proteins from unrelated proteins. However, distinguishing such highly conserved protein family members from other related proteins may not be trivial. I assume, of course, that this is not the problem that the authors are seeking to address, even though it too is an important problem when seeking to assign specific functions to hypothetical proteins. I suggest that the authors describe very precisely the nature of the problem that their new programs aim to address. This appears to be the problem of identifying very distant homologs to the more typical members of a protein superfamily. If so, then the families that fail to be split either by the Cluster algorithm or by the Blue/Cobalt algorithms may be ones that are irrelevant to the problem being addressed because these correspond to highly conserved Pfam families.

Comment 4: I suggest that the authors may get a better perspective on these questions by generating simulated sequences from a known distribution (e.g., based on an HMM) and then adding additional sequences by simulating their evolution, which would model the lack of statistical independence that their programs aim to address.

Minor comments:

1. In Figure 3, what do the gray bars above the colored bars that fade out at their top ends indicate?

2. Page 15 line 209: “brought closer the ROC curve”  “brought closer to the ROC curve”?

**Have the authors made all data and (if applicable) computational code underlying the findings in their manuscript fully available?**

Reviewer #1: Yes

Reviewer #2: Yes

Reviewer #3: Yes

Reviewer #4: Yes

PLOS authors have the option to publish the peer review history of their article (what does this mean?). If published, this will include your full peer review and any attached files.

Reviewer #1: No

Reviewer #2: No

Reviewer #3: No

Reviewer #4: No
---

## [Decision Letter · Decision Letter 1]

26 Jan 2022

Dear Dr. Petti,

Thank you very much for submitting your manuscript "Constructing benchmark test sets for biological sequence analysis using independent set algorithms" for consideration at PLOS Computational Biology. As with all papers reviewed by the journal, your manuscript was reviewed by members of the editorial board and by several independent reviewers. The reviewers appreciated the attention to an important topic. Based on the reviews, we are likely to accept this manuscript for publication, providing that you modify the manuscript according to the review recommendations.

Sincerely,

Maricel G Kann

Associate Editor

PLOS Computational Biology

Feilim Mac Gabhann

Editor-in-Chief

PLOS Computational Biology

[LINK]

Reviewer's Responses to Questions

**Comments to the Authors:**

Reviewer #2: The revision addresses the questions I raised, and I appreciate the discussion in the response to review in particular. This has greatly improved the paper.

Reviewer #3: Sorry if I wasn't clear enough last time. I still feel that much of

the writing about "independent" data is confusing, ambiguous, and

misleading. I think there are two different reasons for this.

Reason (1): Independence is relative, not absolute. According to the

mathematical definition, two things A and B are independent when:

P(A|B) = P(A).

But this depends on the "background" probability distribution, P(A).

Consider these two background probability distributions for sequences:

(i) Random i.i.d. sequences, with some length distribution.

(ii) Randomly pick from a set of real biological sequences, which are

related by evolution.

It is possible for sequences to be independent relative to (ii) but

not (i).

Reason (2): While the title and abstract do not specify "homolgy

search" (and indeed the methods may be useful for other things, like

protein localization prediction), the paper focuses on the aim of

homology search. This is confusing, because "homology" means "related

by evolution", which is the same as the non-independence "nuisance"

that they are trying to eliminate!

In short, I feel that the abstract and introduction should be

carefully rewritten to clarify these issues. Surely the authors

understand all this, and it's well-described in the final paragraph of

Discussion. But the Abstract and Introduction are too unclear.

# MINOR

The paper should at least mention the issue of low-complexity/simple

repeats, and that it might affect the relative performances of the

homology search methods. (I suspect DIAMOND may have an advantage in

avoiding simple repeats.)

Page 20: should "the known sequence x" be "the known sequences x"?

Page 20 last line: x and y mixed up?

Reviewer #4: nothing to upload

**Have the authors made all data and (if applicable) computational code underlying the findings in their manuscript fully available?**

Reviewer #2: Yes

Reviewer #3: Yes

Reviewer #4: Yes

PLOS authors have the option to publish the peer review history of their article (what does this mean?). If published, this will include your full peer review and any attached files.

Reviewer #2: No

Reviewer #3: No

Reviewer #4: No

Figure Files:

Data Requirements:

Reproducibility:

References:

---

## [Editor Report · Decision Letter 2]

10 Feb 2022

Dear Dr. Petti,

We are pleased to inform you that your manuscript 'Constructing benchmark test sets for biological sequence analysis using independent set algorithms' has been provisionally accepted for publication in PLOS Computational Biology.

Best regards,

Maricel G Kann

Associate Editor

PLOS Computational Biology

Feilim Mac Gabhann

Editor-in-Chief

PLOS Computational Biology

---

## [Editor Report · Acceptance letter]

24 Feb 2022

PCOMPBIOL-D-21-01725R2 

Constructing benchmark test sets for biological sequence analysis using independent set algorithms

Dear Dr Eddy,

I am pleased to inform you that your manuscript has been formally accepted for publication in PLOS Computational Biology. Your manuscript is now with our production department and you will be notified of the publication date in due course.

With kind regards,

Anita Estes
